# The Inflammasome Activity of NLRP3 Is Independent of NEK7 in HEK293 Cells Co-Expressing ASC

**DOI:** 10.3390/ijms231810269

**Published:** 2022-09-07

**Authors:** Dominik Alexander Machtens, Ian Philipp Bresch, Jan Eberhage, Thomas Frank Reubold, Susanne Eschenburg

**Affiliations:** 1Institute for Biophysical Chemistry, Hannover Medical School, Carl-Neuberg-Straße 1, 30625 Hannover, Germany; 2Department of Gastroenterology, Hepatology and Endocrinology, Hannover Medical School, Carl-Neuberg-Straße 1, 30625 Hannover, Germany; 3Cluster of Excellence RESIST (EXC 2155), Hannover Medical School, Carl-Neuberg-Straße 1, 30625 Hannover, Germany

**Keywords:** NLRP3, HEK293 cells, inflammasome, NEK7

## Abstract

The cytosolic immune receptor NLRP3 (nucleotide-binding domain, leucine-rich repeat (LRR), and pyrin domain (PYD)-containing protein 3) oligomerizes into the core of a supramolecular complex termed inflammasome in response to microbes and danger signals. It is thought that NLRP3 has to bind NEK7 (NIMA (never in mitosis gene a)-related kinase 7) to form a functional inflammasome core that induces the polymerization of the adaptor protein ASC (Apoptosis-associated speck-like protein containing a CARD (caspase recruitment domain)), which is a hallmark for NLRP3 activity. We reconstituted the NLRP3 inflammasome activity in modified HEK293 (human embryonic kidney 293) cells and showed that the ASC speck polymerization is independent of NEK7 in the context of this cell system. Probing the interfaces observed in the different, existing structural models of NLRP3 oligomers, we present evidence that the NEK7-independent, constitutively active NLRP3 inflammasome core in HEK293 cells may resemble a stacked-torus-like hexamer seen for NLRP3 lacking its PYD (pyrin domain).

## 1. Introduction

The cytosolic receptor NLRP3 (nucleotide-binding domain, leucine-rich repeat (LRR), and pyrin domain (PYD)-containing protein 3) is a key player in the innate immune system. In response to invading pathogens or cellular stress, NLRP3 mounts an inflammatory reaction in the surrounding tissues. The dysfunction of NLRP3 is associated with severe autoinflammatory diseases, which is why NLRP3 emerged as an attractive drug target [1]. NLRP3 is activated by diverse stimuli, such as the bacterial toxin nigericin, viral nucleic acids, extracellular ATP, amyloid-β fibrils, urate or cholesterol crystals, and oxidized mitochondrial DNA [2,3,4,5,6,7]. These very diverse stimuli are sensed upstream of NLRP3 and the resulting signals converge on NLRP3 as the central hub of the activation process. The NLRP3 oligomerizes to form the core of a so-called inflammasome. The PYDs (pyrin domains) of NLRP3 recruit the adapter protein ASC (apoptosis-associated speck-like protein containing a CARD (caspase recruitment domain)) to the inflammasome and thus spark the polymerization of ASC. The ASC filaments aggregate into a single macromolecular ASC-speck that serves as activation platform for pro-caspase-1 autoproteolysis [8,9,10]. Caspase-1 processes the pro-inflammatory cytokines, interleukin-1β (IL-1β) and interleukin-18 (IL-18), into their active forms and activates gasdermin D, which prompts cytokine release and an inflammatory cell death called pyroptosis [2].

The assembly of NLRP3 into the core of the inflammasome is thought to strictly depend on NEK7 (NIMA (never in mitosis gene a)-related kinase 7) [11,12,13]. NEK7 is involved in the formation of the microtubule-based mitotic spindle [14,15] and is supposed to limit NLRP3 inflammasome activity to the interphase of the cell [12]. Based on a cryo-EM (cryo-electron microscopy) structure of NLRP3 lacking the PYD (NLRP3ΔPYD) liganded with NEK7, a disk-shaped undecamer of NLRP3 was modeled using the NLRC4-NAIP (NLR family CARD domain-containing protein 4 — NLR family of apoptosis inhibitory protein) inflammasome core as template [16]. In this model of the NLRP3 inflammasome core, NEK7 bridges the neighboring NLRP3 protomers and thus is an integral component of the disk. However, the disk-like arrangement was not yet confirmed by experimental structure determination. Using cryo-EM, principally different oligomers devoid of NEK7 were reconstructed. These are cage-like oligomers of ten (PDB ID 7PZC) or twelve NLRP3 protomers (PDB IDs 7LFH and 7VTQ) or stacked-torus-like hexamers of NLRP3ΔPYD (PDB IDs 7VTP and 7ZGU). The dodecameric cage-like assembly of NLRP3 was shown to be inactive in terms of ASC polymerization [17], the decameric cage-like assembly of NLRP3 and the stacked-torus-like hexamer were assumed to be inactive [18,19].

To assay the assembly and activation of the NLRP3 inflammasome, numerous different cell-based systems were used. Among the most common systems used are the immune cell lines, such as iBMDMs (immortalized bone-marrow-derived macrophages), the human leukemia monocytic cell line THP-1, or the pro-monocytic, human histiocytic lymphoma cell line U937 [20], which contain all of the components required for the inflammatory reaction towards the activation of IL-1β and IL-18. On the other hand, minimal systems, such as HEK293 (human embryonic kidney 293) or HEK293T (human embryonic kidney 293T) cell lines, which express neither the inflammasome components NLRP3 or ASC nor the interleukins, are frequently used to limit the players to those transiently transfected or genomically integrated into the cells beforehand [21,22,23]. Since all of these cells endogenously contain NEK7, a general requirement of NEK7 for NLRP3 activity was not questioned.

In our study we show that in the HEK293 cells co-expressing ASC, a constitutively active NLRP3 inflammasome core may form that is independent of NEK7. Moreover, we present evidence that this NEK7-independent, active NLRP3 oligomer assembles in a fashion similar to the stacked-torus-like hexamer described for NLRP3ΔPYD.

## 2. Results

### 2.1. NLRP3 Is Constitutively Active in HEK293-ASC Cells Independent of NEK7

We established a cell-based ASC polymerization assay to probe NRLP3 activity in HEK293 cells. The HEK293 cells and HEK293T cells were previously used to characterize inflammasome systems [23]. The cells do not inherently express the inflammasome proteins and thus provide a simple framework to study inflammasome activity, since all of components of the inflammasome are introduced into the system. Although HEK293 and HEK293T cells are easily transfected, we wanted to reduce the unwanted effects that may stem from transfecting multiple plasmids. Therefore, we integrated the gene for the inducible expression of Myc-tagged ASC into the genome of HEK293 cells. We transiently transfected the resulting HEK293-ASC cells with NLRP3 (Appendix A). In the presence of tetracycline, which induces ASC expression in HEK293-ASC cells, the transfection of wild-type NLRP3 readily leads to the formation of ASC specks (Figure 1A; Appendix A). About 70% of the cells successfully transfected with NLRP3 showed ASC specks. The activity of NLRP3 is not increased by stimulation of the cells with nigericin or LPS (lipopolysaccharide) and nigericin (Figure 1B). On the other hand, incubation with the NLRP3 inhibitor MCC950 (also known as CRID3) does not inhibit the activity of NLRP3. If MCC950 is added at the time of NLRP3 transfection or, at a later time point, upon induction of ASC, the fraction of cells containing ASC specks is not decreased (Figure 1B). If NLRP3ΔPYD is transfected, ASC speck formation is abolished as expected (Figure 1B). In order to further validate our assay, we analyzed the effect of the mutation S5D on NLRP3 activity. S5 was previously identified as a phosphorylation site and NLRP3-S5D was shown to be significantly less active towards the initiation of ASC polymerization [24]. Indeed, in our assay the number of cells containing ASC specks was greatly reduced if NLRP3-S5D was transfected instead of wild-type NLRP3 (Figure 1B; Appendix A).

While HEK293 cells do not contain canonical inflammasomal components, the cells endogenously express NEK7 [25,26]. Besides the role of NEK7 in cell division [15], the kinase was shown to be indispensable for the activity of NLRP3 in iBMDMs primed with LPS and nigericin [12,16,27]. To assay the dependency of the activity of NLRP3 on NEK7 in HEK293-ASC cells, we introduced the triple mutation S796A/Q798A/K799A into NLRP3. The triple mutation was shown to disturb the binding of NEK7 to NLRP3 and to thus abrogate NLRP3 activity in iBMDMs [16,28]. If NLRP3-S796A/Q798A/K799A was transfected into HEK293-ASC cells, the observed activity of the mutant NLRP3 was, surprisingly, as high as that of the wild-type protein (Figure 1C). To confirm this unexpected observation, we knocked out NEK7 in HEK293-ASC cells (Appendix A). In the NEK7 knockout cells, wild-type NLRP3 and the mutant NLRP3-S796A/Q798A/K799A are fully active (Figure 1C). Apparently, under the conditions used in our assay, NLRP3 forms an active species that functions independently of NEK7.

### 2.2. Mutations in the Interfaces of the NLRP3 Hexamer Influence NLRP3 Activity

To seek a structural basis for the stimulation-independent activity observed in HEK293-ASC cells, we examined the NLRP3 oligomers known so far. Since the activity that we observed in our HEK293-ASC cells is independent of NEK7, we excluded the modelled disk-like undecamer [16], which strictly depends on NEK7. The NEK7-independent cage-like decamer (PDB ID 7PZC), cage-like dodecamer (PDB IDs 7LFH and 7VTQ), and stacked-torus-like hexamer (PDB IDs 7PGU and 7VTP) are each built from NLRP3 protomers in the closed conformation (Appendix A). The full-length NLRP3 consists of a pyrin domain (PYD), a nucleotide binding and oligomerization domain (NOD), and a leucine-rich repeat domain (LRR) (Appendix A). The NOD is subdivided in the nucleotide binding domain (NBD), the helical domains 1 and 2 (HD1 and HD2), and the winged-helix domain (WHD).

The cage-like and hexameric NLRP3 oligomers have a distinct LRR–LRR interface in common (Figure 2), which was dubbed “back-to-back” interface in the cage-like dodecamer [17]. If the back-to-back interface is disturbed by mutations, the cage-like NLRP3 dodecamer is disrupted [17]. If we disturb the prominent interactions in the back-to-back interface (Figure 2) in our HEK293-based assay, using mutant NLRP3 that carries the mutations D789R, R816D, or F788E/F813E, the ability of mutant NLRP3 to initiate ASC specks is only marginally reduced compared to that of wild-type NLRP3 (Figure 3A).

A closer inspection of the structures of the stacked-torus-like hexamer (PDB IDs 7PGU and 7VTQ) reveals that the mutations in the back-to-back interface are likely to hamper the attachment of the two trimeric rings to each other, preventing the formation of the hexamer. To follow up on this idea we recombinantly produced human NLRP3 lacking the PYD (NLRP3ΔPYD, residues 126-1036) and NLRP3ΔPYD carrying the double mutation F788E/F813E (NLRP3ΔPYD-F788E/F813E) in Sf9 insect cells and purified the proteins to homogeneity. We then analyzed the oligomeric states of NLRP3ΔPYD and NLRP3ΔPYD-F788E/F813E by size exclusion chromatography (SEC) (Figure 3B) and dynamic light scattering (DLS) (Figure 3C). We indeed see that the oligomeric species of the mutant is smaller than that of the wild-type NLRP3ΔPYD. The elution volume in the SEC experiment as well as the hydrodynamic radius in the DLS experiments are well in line with the magnitudes expected for an NLRP3ΔPYD trimer.

The two identical trimeric rings of the stacked-torus-like hexamer are stabilized by interactions between the NOD and the LRR of the adjacent NLRP3 protomers (Figure 3D). This NOD–LRR interface is unique to the hexameric assembly of NLRP3. If we disturb the NOD–LRR interface (Figure 3D) by the mutation Y143R, the activity of mutant NLRP3 is reduced to only 60% of the activity of wild-type NLRP3 (Figure 3A). If we combine the mutations of the LRR–LRR interface and of the NOD–LRR interface, as in NLRP3-Y143R/D789R or NLRP3-Y143R/F788E/F813E, the speck formation is reduced to less than 40% (Figure 3A). The key residue Y143 is part of an element that has been dubbed “polybasic linker” (residues 131–147). The “polybasic linker” was postulated to mediate binding to negatively charged lipids on the trans-Golgi membrane [29]. In a recent publication, the positively charged side chains in this element were shown to be required for membrane binding of NLRP3 upstream of the inflammasome formation in iBMDMs [17]. To analyze whether the significant reduction in NLRP3 activity in HEK293-ASC cells by the mutation Y143R, which resides in this linker helix, results from disturbance of the described membrane binding steps, we mutated the other residues of the helix and transfected NLRP3-K142S, NLRP3-R145A, or NLRP3-R147S in HEK293-ASC cells. The activities of the mutant proteins are comparable to the activity of wild-type NLRP3 (Figure 3A).

## 3. Discussion

We demonstrated that NLRP3 is able to induce the polymerization of ASC in HEK293-ASC cells without the requirement of NEK7. Our results expand the view that NEK7 is essential for the formation and activity of the NLRP3 inflammasome [12,16,17,27,30]. The mutations in NLRP3 that abrogate NEK7 binding [16], as well as the knock-out of endogenous NEK7 in HEK293-ASC cells, do not influence the observed activity of NLRP3 as detected by ASC speck formation. This suggests that an alternative active NLRP3 oligomer species exists that is substantially different from the NEK7-containing disc-like inflammasome core [16]. Moreover, this alternative NLRP3 inflammasome core is insensitive towards the well-established inhibitor MCC950 (Figure 1B), which, again, distinguishes the alternative species from the NEK7-containing disc-like inflammasome. MCC950 is thought to fix NLRP3 in its closed conformation [18,31], thus preventing the assembly of an NAIP/NLRC4-like NLRP3 inflammasome that relies on opening of the NLRP3 protomers [16].

It is tempting to assume that an active NEK7-independent NLRP3 oligomer adopts the hexameric arrangement seen in the NLRP3ΔPYD structures. This notion is supported by the fact that disturbing the NOD–LRR interface unique to the NLRP3ΔPYD hexamer by selected point mutations substantially reduces the activity of NLRP3 in HEK293-ASC cells (Figure 3A). Interestingly, an NLRP3 hexamer was detected earlier as the activator of ASC fibrillation in a cell-free expression system using single-molecule fluorescence techniques [32]. In full-length NLRP3, the PYDs may protrude from each side of the stacked-torus-like hexamer to self-assemble into trimers that serve as seeds for ASC polymerization (Figure 4).

It is not clear yet how many PYDs have to initially oligomerize to function as seed for ASC polymerization. It was shown previously that expression of the PYD of NLRP3 fused to a trimerization domain is sufficient to induce IL-1β secretion and pyroptosis in iBMDMs [33]. Moreover, inspection of the geometry of the ASC–PYDs in the cryo-EM reconstruction of an ASC–PYD filament [10] suggests that the free interaction sites in an initial PYD trimer would properly direct the assembly of the ASC filament. The assumption that a trimeric platform is sufficient for inducing ASC polymerization is in line with our finding that NLRP3 carrying the mutation F788E/F813E was only marginally less active than wild-type NLRP3 (Figure 3A). This mutation only affects the LRR–LRR interface and may lead to dissociation of the hexamer into two trimeric rings (Figure 2 and Figure 3C). If, on the other hand, the intra-trimer interface seen in the NLRP3ΔPYD hexamer is compromised by mutations, the activity of NLRP3 in HEK293-ASC cells is substantially reduced (Figure 3A).

In the NLRP3 hexamer, the polybasic-linker helix at the N-terminal end of the NOD is an integral part of the NOD–LRR interface (Figure 3D). The movement of the linker helix, however, is thought to be the molecular event that leads to activation of the inflammasome in response to the reduction of the intracellular potassium concentration induced by nigericin [34]. The finding that the activity of NLRP3 is independent of stimulation by nigericin in our HEK293-ASC cell assay (Figure 1B), which was previously observed in HEK293(T) cells [24,35], corresponds well with a hexameric arrangement of NLRP3, where the linker helix is fixed in its resting position. In the geometry of the NLRP3 hexamer, the length of the unstructured linker between the PYD and the polybasic-linker helix is sufficient to properly arrange the PYDs on top of the trimeric rings without moving the linker helix.

It is unclear how a hexameric assembly may form if NLRP3 is in full-length. When full-length NLRP3 was overexpressed in insect cells or in HEK293T cells, the protein preferentially assembled in cage-like structures of 10-, 12-, 14-, and 16-mers of NLRP3 with the PYDs sequestered in the lumen of the cage [17,18]. The decisive event might be the position of the PYD relative to the body of NLRP3. If the PYD is close to the NOD or LRR of NLRP3, the hexamer cannot be formed and the cage-like storage form is favored. If the PYD is remote of the body of NLRP3, the hexamer may assemble. Such a speculative scenario is in line with the finding that deletion of the PYD prevents the formation of NLRP3 cages [17] and leads to the formation of the hexameric complex [19]. However, given the high degree of regulation of NLRP3 activation in immune cells, it is unlikely that the hexameric inflammasome core of NLRP3 is built under these physiological conditions. The HEK293 and HEK293T cell lines on the other hand do not endogenously provide the environment for canonical NLRP3 signaling and therefore are likely to permit the formation of an alternative NLRP3 inflammasome species, which is constitutively active. We have demonstrated that such a constitutively active NLRP3 species indeed exists in HEK293-ASC cells and that this alternative inflammasome species is the predominant initiator of ASC polymerization in these cells. In about 70% of the cells that were transfected with NLRP3, the ASC specks readily form after induction of ASC expression. Neither stimulation with nigericin nor the absence of NEK7 nor the inhibitor MCC950 affect the activity of NLRP3 in the HEK293-ASC background. In this light, the outcome of the studies that employed HEK293(T) cells to examine NLRP3 inflammasome biology and to assess the influence of exogenous agents on the activity of NLRP3 have to be carefully re-evaluated. The reconstitution of NLRP3 inflammasome activity in the minimal system of HEK293(T) cells may build on molecular determinants that differ substantially from those in the immune cell lines.

## 4. Materials and Methods

### 4.1. Cloning, Expression and Purification of Recombinant Human NLRP3

The coding sequence corresponding to the N-terminally truncated human NLRP3ΔPYD (amino acids 126-1036) fused to a short C-terminal Linker (GSGG) and a FLAG tag (DYKDDDDK) was cloned into the pFastBac Dual (Invitrogen, Waltham, MA, USA) baculovirus transfer vector under control of the polyhedrin promoter. The mutant constructs were generated by megaprimer mutagenesis. A recombinant baculovirus was produced using the MultiBac system (Geneva Biotech, Pregny-Chambésy, Switzerland) following the manufacturer’s protocols. For the protein expression in *Sf*9 (*Spodoptera frugiperda*) cells, 1.8 × 10^6^ cells/mL were infected with recombinant baculovirus and harvested 72 h post-infection. The cells were lysed by sonication in lysis buffer containing 50 mM Tris-HCl pH 8.0, 140 mM NaCl, 1 mM MgCl_2_, 0.5 mM DTT, 0.2% (*v*/*v*) Nonidet P-40 and 1 mM PMSF. After centrifugation, the supernatant was applied to an Anti-FLAG M2 Affinity Agarose Gel column (Sigma Aldrich, St. Louis, MO, USA). The bound FLAG-tagged NLPR3-ΔPYD protein was eluted with 0.1 mg/mL FLAG-peptide (Sigma Aldrich), concentrated using Amicon ultrafiltration devices (Merck, Darmstadt, Germany), and further purified by size exclusion chromatography on a Superose 6 Increase 10/300 column (Cytiva, Marlborough, MA, USA) equilibrated with a buffer containing 20 mM HEPES pH 7.5, 200 mM NaCl, 1 mM MgCl_2_, and 2 mM DTT. All of the purification steps were performed at 4 °C.

### 4.2. Analytical Size Exclusion Chromatography

A total of 100 µL 1 mg/mL FLAG-tagged NLRP3ΔPYD (wild-type or mutant) were applied to a Superose 6 Increase 10/300 column (Cytiva) equilibrated with a buffer containing 20 mM HEPES pH 7.5, 200 mM NaCl, 1 mM MgCl_2_, and 2 mM DTT at a flow rate of 0.3 mL/min. The molecular weight was estimated by comparing the elution volume with a set of calibration standards (5 mg/mL Thyreoglobulin (669 kDa), 0.3 mg/mL Ferritin (440 kDa), 4 mg/mL Aldolase (158 kDa), 4 mg/mL Ovalbumin (44 kDa)).

### 4.3. Dynamic Light Scattering

The particle size distributions of purified NLRP3ΔPYD were analyzed in a Viscotek dynamic light scattering (DLS) system (Malvern Instruments, Malvern, UK). All of the measurements were performed in triplicates at 10 °C in a buffer containing 20 mM HEPES pH 7.5, 200 mM NaCl, 1 mM MgCl_2_, and 2 mM DTT. The autocorrelation curves from a set of 20 acquisitions with 20 s integration time each were calculated using the Omnisize 3.0 software (Malvern Instruments).

### 4.4. ASC Speck Assay

For the functional reconstruction of speck formation in the HEK293 background, we generated a cell line that stably expressed C-terminally Myc-tagged ASC with a tetracycline inducible promotor, using Flp-In TREx technology (Invitrogen) according to the manufacturer’s instructions (HEK293-ASC). The C-terminally Myc-tagged human ASC was cloned into the pcDNA5/FRT/TO vector and full-length human FLAG-tagged NLRP3 and the mutant constructs were cloned into the mammalian expression vector pcDNA3.1. All of the cell lines were regularly tested to be mycoplasma-negative by PCR.

The cells were seeded onto 15 mm coverslips that were coated with poly-l-lysine (Sigma Aldrich) and transfected with 500 ng of the indicated NLRP3 constructs using the Lipofectamine 2000 system (Thermo Fisher Scientific, Waltham, MA, USA). After 18 h, ASC expression was induced by adding 0.2 ng/µL tetracycline. Then, 24 h later, the samples were washed twice with DMEM (Dulbecco’s modified Eagle’s medium), fixed for 30 min with 1% paraformaldehyde, washed twice with PBS, permeabilized for 5 min with 0.1% Triton X-100, washed twice with PBS and blocked for 1 h in 2% BSA in PBS (blocking solution). Afterwards, primary antibodies against M2-Flag (Sigma Aldrich: F1804) and Myc (Cell Signaling: 71D10) were diluted 1:100 in blocking solution and applied for 45 min at room temperature. The samples were washed twice in PBS and incubated for 1 h in solutions containing the appropriate secondary antibodies (Goat-anti-rabbit-AF488, Jackson Immuno: 111-545-144; Goat-anti-Mice-Cy3, Jackson-Immuno: 115-165-146) diluted 1:400 in blocking solution. The coverslips were washed twice with PBS and ddH_2_O and mounted with Imunoselect antifading mounting-medium (Dianova, Hamburg, Germany).

Images were collected with a Leica TCS SP8 Confocal microscope using a 63×/1.4 NA objective. The NLRP3 activity was determined by calculating the percentage of ASC speck-positive cells divided by the total number of transfected cells. The image analysis was completed manually using Fiji software and the Cell Counter plugin [36]. A minimum of six images with a total of at least 100 cells was analyzed in each experiment. The entire experiment was independently repeated at least three times. Brightness and contrast were adjusted for better visibility.

### 4.5. CRISPR/Cas Knockout of NEK7 in HEK293-ASC Cells

HEK293-ASC cells were lipofected with PX459 [37] (Addgene #62988 provided by Jens Ingo Hein and Jan Faix) containing the guide sequence AAGGCCTTACGACCGGATAT against NEK7 Exon 3. The transfected cells were selected by subsequent 2.5 µg/mL puromycin treatment for 4 days with a daily medium change. The remaining cells were diluted to 1 cell/µL and divided onto several 96-well plates in 1 µL drops. DMEM was added to wells containing a single cell which was cultured to confluence for gDNA extraction over several weeks. The gDNA was extracted (11796828001, Roche, Basel, Switzerland) and amplified using NEK 7 Exon 3 flanking primers (AGTTCAGACTCTTCATCCCTATGT and CAACGTTCAATCTTTCCCAGCA) and Phusion Polymerase (F530S, ThermoFisher) and sequenced by Microsynth Seqlab GmbH, Göttingen, Germany. The sequencing results were evaluated using the TIDE webserver [38]. The clones with a homozygous NEK7 knockout were expanded further for Western blot analysis. The cell lysate of a 10 cm cell culture dish was prepared using RIPA buffer and subjected to SDS-PAGE. Semidry blotting to a nitrocellulose membrane was performed for 1 h with 13 V followed by 1 h of blocking with 5% Milk in TBST. The blot was subjected to 1:5000 Rabbit-anti-NEK7 (abcam ab133514) in Milk/TBST for 16 h at 4 °C and washed with TBST. Incubation with 1:1000 HRP-conjugated anti-Rabbit antibody (ThermoFisher #32460) in Milk/TBST for 3 h was performed thereafter. NEK7 was imaged using ECL substrate (ThermoFisher #34095) after thorough washing. Ponceau coloration was performed to confirm the successful loading and blotting procedures.

## Figures and Tables

**Figure 1 ijms-23-10269-f001:**
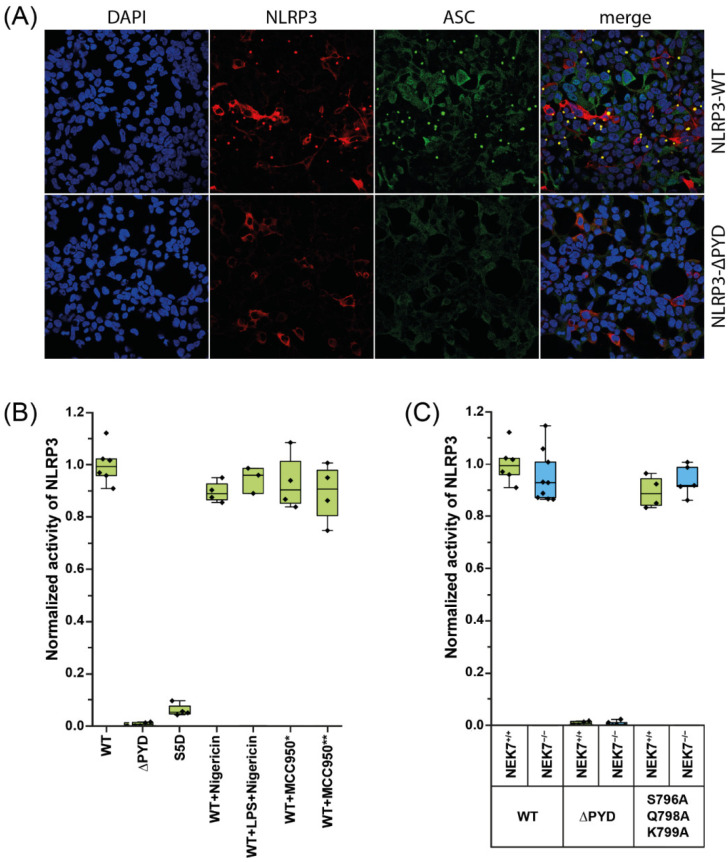
Quantification of ASC speck containing HEK293-ASC cells after transfection of wild-type or mutant NLRP3. (**A**) Representative confocal fluorescence images of HEK293-ASC cells transfected with wild-type (NLRP3-WT) or PYD-deleted (NLRP3ΔPYD) NLRP3-Flag. Primary antibodies against M2-Flag and Myc were used to detect NLRP3-Flag (red) and ASC-Myc (green). Nuclei were stained with DAPI (blue); (**B**) Quantification of HEK293-ASC cells containing ASC specks after transfection of wild-type or mutant NLRP3 in the presence or absence of nigericin, LPS, and MCC950. MCC950 was added during and after ASC expression induction (*) or during and after NLRP3 transfection (**). Box plots represent the ratio of speck-positive cells to the total of transfected cells. They show the interquartile range (IQR) and the median. Each dot represents one experiment. Whiskers include experiments within 1.5× IQR. For each experiment, six images were counted with at least one hundred cells each; (**C**) Quantification of wild-type or NEK7-knockout HEK293-ASC cells containing ASC specks after transfection of wild-type or mutant NLRP3. Box plots represent the ratio of speck-positive cells to the total of transfected cells. They show the IQR and the median. Each dot represents one experiment. Whiskers include experiments within 1.5× IQR. For each experiment, six images were counted with at least one hundred cells each.

**Figure 2 ijms-23-10269-f002:**
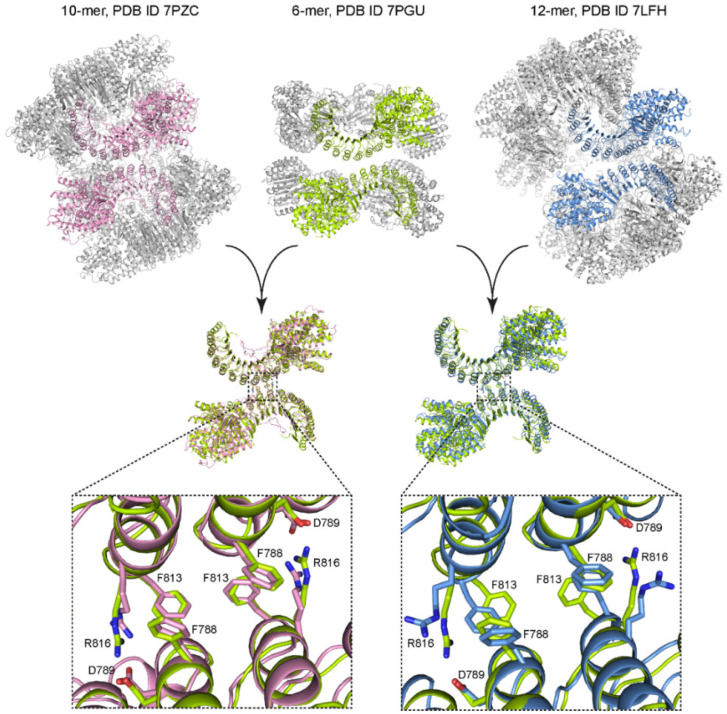
Comparison of experimental structures of complexes of NLRP3. Shown in the top row are cartoon representations of decameric (**left**) and dodecameric (**right**) complexes of full-length NLRP3 as well as of a hexameric complex of PYD-deleted NLRP3. Superpositions of back-to-back dimers from the decamer and the dodecamer with the hexamer (as indicated by the arrows) are shown in the middle row. The bottom row shows magnified views on the dimer interfaces from the superpositions. Amino acids involved in the interface are shown in stick representation and are labelled according to the sequence numbering of human NLRP3.

**Figure 3 ijms-23-10269-f003:**
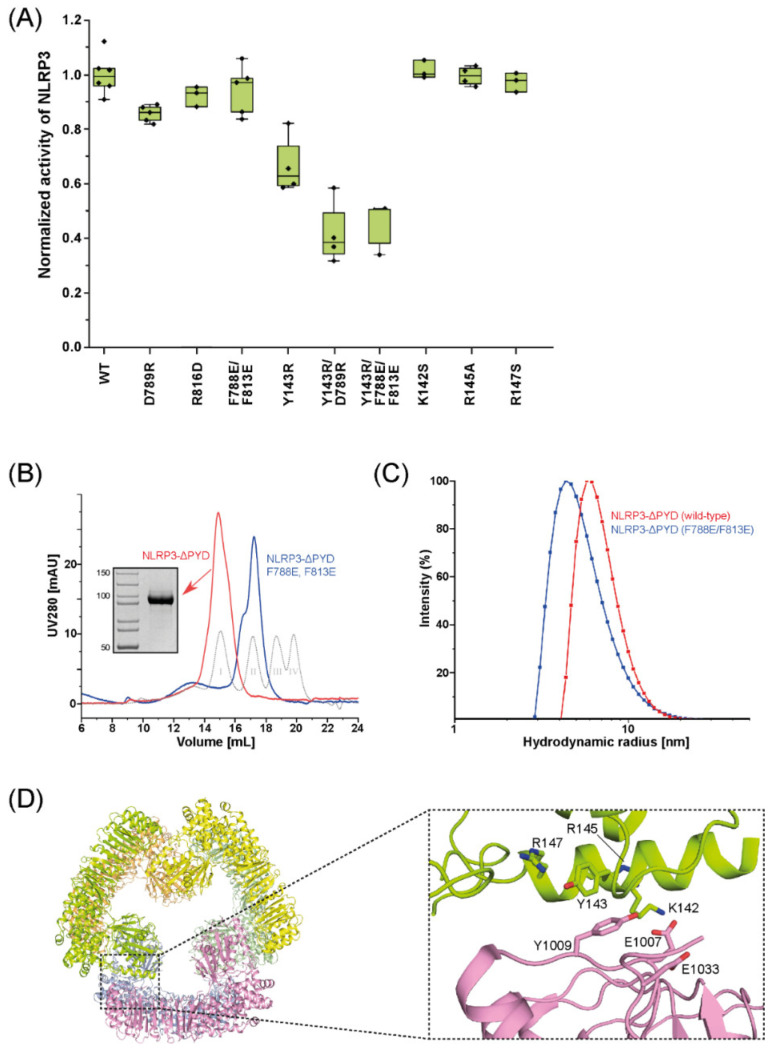
Characterization of NLRP3 mutants. (**A**) Quantification of HEK293-ASC cells containing ASC specks after transfection of wild-type or mutant NLRP3. Box plots represent the ratio of speck-positive cells to the total of transfected cells. They show the interquartile range (IQR) and the median. Each dot represents one experiment. Whiskers include experiments within 1.5× IQR. For each experiment, six images were counted with at least one hundred cells each; (**B**) Size exclusion chromatography of wild-type NLRP3ΔPYD (red) and NLRP3ΔPYD-F788E/F813E (blue). Peaks corresponding to marker proteins (I: Thyroglobulin, 669 kDa; II: Ferritin, 440 kDa; III: Aldolase, 158 kDa; IV: Ovalbumin, 44 kDa) are shown as gray dotted lines; (**C**) Dynamic light scattering of wild-type NLRP3ΔPYD (red) and NLRP3ΔPYD-F788E/F813E (blue). The peak representing the mutant is shifted to a smaller hydrodynamic radius; (**D**) (**Left**): Cartoon representation of the top view of the NLRPΔPYD hexamer. (**Right**): Close-up view of the NOD–LRR interface. Amino acids involved in the interface are shown in stick representation and are labelled according to the sequence numbering of human NLRP3.

**Figure 4 ijms-23-10269-f004:**
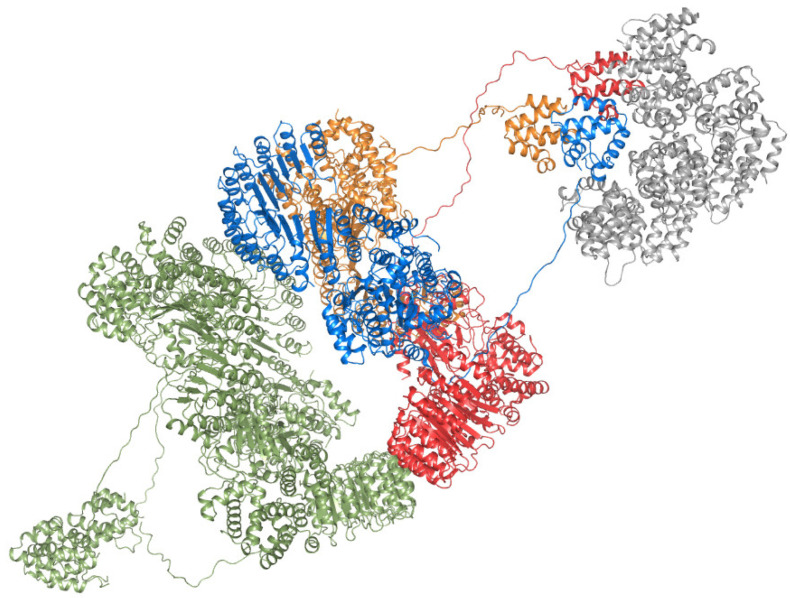
Model of NLRP3-mediated generation of ASC filaments. A hypothetical hexamer of full-length NLRP3 is shown in cartoon representation with the protomers of one trimeric ring colored in orange, red, and blue, respectively. The pyrin domains have been attached to the N-termini of the protomers of the PYD-deleted hexamer (PDB ID 7PGU) via their flexible linkers in a hypothetical conformation. The ASC-filament is shown in grey.

## Data Availability

Not applicable.

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
