# Peer review of "The Inflammasome Activity of NLRP3 Is Independent of NEK7 in HEK293 Cells Co-Expressing ASC"

_ijms, 2022, doi:10.3390/ijms231810269_

Round 1

Reviewer 1 Report

Manuscript # 1850326

In the current manuscript titled " NLRP3 activity is independent of NEK7 in HEK293 cells co-expressing ASC” the authors have determined the role of NLRP3 in the activation of inflammasomes which is independent of well-established inflammasome complex components such as NIMA (never in mitosis gene a)-related kinase 7 (NEK7). Authors have established the structural basis of the NLRP3-ASC component in inflammasome complex formation which is independent of NEK7 involvement. AS inflammasome activation is an important inflammatory cell death mechanism associated with the innate immune system and promotes inflammation. Overactivation of the NLRP3-mediated inflammasome complex promotes inflammatory diseases and activation of caspase-1, IL-1β, and IL-18. Thus, understanding the NLRP3-mediated inflammasome complex could be a therapeutic target. However, I have the following comments for this study.

1)    Though canonical NLRP3-mediated inflammasome complex formation and inflammation involve the activation of caspase-1, IL-1β, and IL-18. In the HEK293 system, authors have determined the interaction of the NLRP3-ASC component with NEK7. However, they did not talk about the activation and release of caspase-1, IL-1β, and IL-18. Are NEK7-independent NLRP3 inflammasome work without caspase-1 activation and cytokines release? Authors should perform experiments by taking HEK293 cells and immune cells to see the interaction of NEK7 and NLRP3.

2)    Since canonical NLRP3-mediated inflammasome activation involved phosphorylation of NLRP3. Because the HEK293 cell system does not allow inflammasome formation. Thus, did the authors try to see if NEK7 independent activation of NLRP3 involves phosphorylation of NLRP3?

3)      In figure 1A, the authors have performed immunostaining of NLRP3 and ASC-co-expressing HEK293 cells. Inflammasome formations take place inside cells. But in this study co-staining is occurring outside cells but not inside cells. NLRP3 and ASC staining are occurring in different cells but are not co-localizing. This could be a drawback of this paper. This staining could be background staining. Authors should explain this and show co-localization by an arrowhead and different images.  

4)    Since, in this study neither stimulation with nigericin nor the absence of NEK7 nor the inhibitor MCC950 affect the activity of NLRP3 in the HEK293-ASC background.  Thus, the authors should perform this experiment in both HEK293 and iBMDM to see the difference in NLRP3 activation. Because inflammatory and non-inflammatory cell systems could be different for NLRP3 activation.

5)      Do authors try to see whether NEK7-independent NLRP3 inflammasome formation follows canonical or noncanonical pathways in relation to the NEK7 role?  

Reviewer 2 Report

Authors in this manuscript used HEK293-ASC cells and existing structural models of NLRP3 to show that NLRP3 induces ASC polymerization independently of NEK7 in HEK293 cells and propose a stacked-torus-like hexamer for this NLRP3 structure. specific comments are below. 

1, authors should rephrase the title "NLRP3 activity" to be more specific, such as "NLRP3 induces ASC polymerization...". NLRP3 activity also refers to its ATPase activity.

2, authors should provide expression levels of NLRP3 for figure1 (B and C); and figure 3A. the difference of NLRP3 activity measured by ASC polymerization might be simply due to the protein expression levels. 

3, in the method section, the authors should describe how NEK7 KO cells were generated.

4, authors should comment on previous studies showing that NLRP3 PYD and NLRP3 mutants lacking the LRR lost their interaction with NEK7 but still induced ASC polymerization in HEK293 cells. 

Reviewer 3 Report

NLRP3 activity is independent of NEK7 in HEK293 cells co-expressing ASC

Machtens et. al.

The authors present an examination of the effects of NLRP3 inflammasome mediated formation of ACS filaments under different circumstances in HEK293 cells that have been modified to express myc-tagged ACS.  The authors conclude from their study that NEK7, a kinase thought to influence NLRP3 activity is not required for NLRP3 activity in HEK293 cells.  In addition, the authors show that several interactions between amino acid residues responsible for forming hexameric structures of NLRP3 are essential for full activity of the complex.  The major strength of this study is the data showing that NEK7, which is assumed to be integral to NLRP3 activity may not be essential as previously suspected.  The major weakness of the study is the lack of data, as a more comprehensive histological and IP profile would provide further evidence for NEK7 being unnecessary for ACS filament formation.  The conclusions of the study may therefore not be entirely appropriate before additional experiments are performed.  Several specific suggestions for improvement are given below.

The introduction contains a large number of acronyms, which present a large amount of jargon. However, I commend the authors on doing a nice job of sufficiently defining the acronyms at their first occurrence in this section.

Line 98 should contain a reference to figures 1A&B, since the images show a striking reduction in ASC specs with the S5D mutant.

Line 121 – Are there no stained images of the S796A/Q798A/K799A mutant transfected cells?  This should be shown, especially in the absence of NEK7.

Line 154 – again, it would be helpful to see these images and not just the quantification.

Round 2

Reviewer 1 Report

The Authors made significant changes and incorporated the reviewer's suggestions. 

Reviewer 3 Report

The addition of the requested images greatly enhances the quality of the manuscript.  Following minor modifications to the grammatical format of the manuscript, it is my opinion that it will be suitable for publication.